# Changed Detection Based on Patch Robust Principal Component Analysis

**Wenqi Zhu** [1,†] , **Zili Zhang** [2,†] , **Xing Zhao** [1] and **Yinghua Fu** [1,*]

1   School of Optical-Electrical and Computer Engineering, University of Shanghai for Science and Technology, Shanghai 200093, China; 1935023630@st.usst.edu.cn (W.Z.); 202440454@st.usst.edu.cn (X.Z.)
2   School of Computer Science and Technology, Huazhong University of Science and Technology, Wuhan 430074, China; U201915148@hust.edu.cn
*   Correspondence: fuyh@usst.edu.cn
†   These authors contributed equally to this work.

**Abstract:** Change detection on retinal fundus image pairs mainly seeks to compare the important differences between a pair of images obtained at two different time points such as in anatomical structures or lesions. Illumination variation usually challenges the change detection methods in many cases. Robust principal component analysis (RPCA) takes intensity normalization and linear interpolation to greatly reduce the illumination variation between the continuous frames and then decomposes the image matrix to obtain the robust background model. The matrix-RPCA can obtain clear change regions, but when there are local bright spots on the image, the background model is vulnerable to illumination, and the change detection results are inaccurate. In this paper, a patch-based RPCA (P-RPCA) is proposed to detect the change of fundus image pairs, where a pair of fundus images is normalized and linearly interpolated to expand a low-rank image sequence; then, images are divided into many patches to obtain an image-patch matrix, and finally, the change regions are obtained by the low-rank decomposition. The proposed method is validated on a set of large lesion image pairs in clinical data. The area under curve (*AUC*) and mean average precision (*mAP*) of the method proposed in this paper are 0.9832 and 0.8641, respectively. For a group of small lesion image pairs with obvious local illumination changes in clinical data, the *AUC* and *mAP* obtained by the P-RPCA method are 0.9893 and 0.9401, respectively. The results show that the P-RPCA method is more robust to local illumination changes than the RPCA method, and has stronger performance in change detection than the RPCA method.

**Keywords:** changed detection; P-RPCA; low-rank decomposition

## 1. Introduction

In recent decades, due to the development of digital image processing system technology, fundus imaging technology has been improved [1]. Digital retinal imaging technology can be used for the clinical diagnosis and management of retinal diseases [2–4]. Detecting changes in a pair of images is one of the most commonly encountered low-level tasks in medical image analysis by which the important change is to be identified between two different time stages from the same scene [5]. The goal of change detection is to identify a significant change and remove the unimportant one such as camera motion, illumination variation and nonuniform attenuation. As the changes between different image pairs are diverse and the important changes vary in different applications [6,7], it makes the change detection methods relatively challenging. Important changes in a retinal fundus image mainly include a change of retinal tissue, anatomical structure or lesions [8,9].

Change detection methods consist of preprocessing, producing and analyzing the difference image [10,11]. The pair of images is registered in the location and adjusted in the intensity to each other at the preprocessing step, then they are compared to generate a difference image, finally the change features are segmented from the difference image.

However, due to the different imaging conditions, it is difficult to remove the illumination variations on image change detection only by preprocessing, which makes it necessary for the detection algorithm to be robust to illumination. Recently, many researchers have used supervised learning methods to detect changes, and although these methods are robust to complex intensity, they requires a large amount of labelled data [12–14]. For medical image pairs, it is impossible to use supervised methods because of the great variations of change regions and a few available training samples.

In the past, most unsupervised change detection algorithms focused on comparing the image pairs pixel wisely such as the method based on image difference or image quotient [11,15,16]. However, nonuniform illumination is very common in retinal imaging [17], and the pixel-by-pixel comparison method is frangible to the illumination variations between the images, which immensely challenges intensity normalization techniques. Many researchers have put great efforts to designing various models to deal with illumination [5]. The iterative robust homomorphic surface fitting (IRHSF) was especially conceived to model the illumination for the fundus image by calculating the curvature of the retinal surface [11], which can correct the intensity of the image well, but the change regions are still distracted obviously by the illumination variation.

Change detection methods based on background modeling have been proposed in recent years [10,18] whose core idea stems from video surveillance [19]. For longitudinal fundus images, the anatomic structures with the illumination together are modeled as the background, and only the interested change regions are kept as the foreground [10,18]. For the fundus image pair, the normal anatomical structures evolve slowly and change little over time, which can be regarded as the background model. In order to obtain an invariant background, the illumination variations between the image pair are filtered as the unimportant change after intensity normalization and linear interpolation, so that they are removed from the background model and change regions as the reconstructed error by low-rank decomposition.

In order to remove the distraction of intensity from the background model and change regions, Fu et al. [10] combined the intra-image correction with the inter-image normalization and linear interpolation to smooth the illumination variations between the fundus image pair. By doing this, the local and global illumination variations were greatly reduced and filtered by decomposing the image matrix. This matrix-RPCA approach can reduce the illumination variations well and detect the clear change regions, but it is still sensitive to the local intensity abruptness such as the random light spots or some local uneven intensity. As the local intensity abruptness exists in the fundus image pair, the detection result of the matrix-RPCA approach will be significantly distracted in many cases and no further solution is mentioned in the literature. Inspired by patch-group-based tensor RPCA (PG-TRPCA) [18,20], we develop a patch-based RPCA (P-RPCA) method to remove the distraction of random light spots in the detection of the changes between a pair of fundus images. The matrix-RPCA method converts each image into a column vector and the entire image sequence into the image matrix. Compared with RPCA, the P-RPCA method divides the spanned image sequence into many subsequences by turning the image into patches after illumination correction and linear interpolation, and then concatenates these subsequences and finally vectorizes the subsequences into the image matrix and decomposes the image matrix to obtain the change regions.

The contributions of this paper can be summarized as follows. First, the P-RPCA is proposed to detect the change regions and remove the random light spots. As the RPCA based on the image matrix is robust to the global illumination variations but leaves the local intensity intact, the P-RPCA can deal with the global and local illumination variations at the same time. Second, the patch-based subsequences are concatenated together and vectorized into an image matrix, which increases the rank of the image matrix and is useful to obtain a more robust background and clear change regions.

The rest of this paper is organized as follows. Section 2 gives the preprocessing techniques which turn a pair of images into an image sequence. Section 3 presents the proposed change detection method in detail. The experiment results and discussion are given in Section 4. This paper will be concluded with some discussions of future work in Section 5.

## 2. Preprocessing

For change detection, the illumination variations between image pairs distract the change regions obtained by many methods. There are still a lot of illumination variations between a pair of images after intensity normalization. Hence, the linear interpolation is used to reduce the illumination variation further between the successive images, which makes the sequence have more low-rank components at the same time.

### 2.1. Intensity Correction

Many studies have presented intensity correction techniques inside an image, such as color normalization [21,22], contrast enhancement [11,23], and nonuniform illumination correction [24]. As shown in Figure 1, the original image taken as the reference image was that from DRIVE. DRIVE was established to enable comparative studies on the segmentation of blood vessels in retinal images and acquired using a Canon CR5 non-mydriatic 3CCD camera with a 45-degree field of view (FOV) [25]. After correcting the intra-intensity of each image, the image pair is adjusted into two different intensity levels as shown in Figure 1a,b. The global illumination variation is still obvious, and hence we take the intensity normalization to correct this kind of illumination variations for the image pair.

Suppose that there are two original images $I_1$ and $I_2$ denoting the reference image and current image, respectively, which will be enhanced by some intensity correction techniques such as IRHSF [11] and denoted by $\tilde{I}_1$ and $\tilde{I}_2$ separately. Then, $\tilde{I}_1$ is adjusted to the intensity of $\tilde{I}_2$ to obtain the normalized images $\hat{I}_{12}$, and $\tilde{I}_2$ is adjusted to the intensity of $\tilde{I}_1$ to obtain the normalized image $\hat{I}_{21}$ [10]. The formulas of $\hat{I}_{12}$ and $\hat{I}_{21}$ are given as follows:

$$\hat{I}_{12} = \frac{\sigma_2}{\sigma_1}\{\tilde{I}_1 - \mu_1\} + \mu_2 \tag{1}$$

and

$$\hat{I}_{21} = \frac{\sigma_1}{\sigma_2}\{\tilde{I}_2 - \mu_2\} + \mu_1 \tag{2}$$

where $\mu_i$ and $\sigma_i$ are the mean value and standard deviation of $\tilde{I}_i$, $i = 1, 2$.

Figure 1 illustrates the result of intensity normalization for a pair of fundus images. In Figure 1, (c) is closer to (b) than (a) for the intensity, and (d) is closer to (a) than (b) for the intensity. The image pair showing in (a) and (b) is adjusted to two image pairs with different intensity levels: a brighter image pair (a) and (d) and a darker image pair (b), (c).

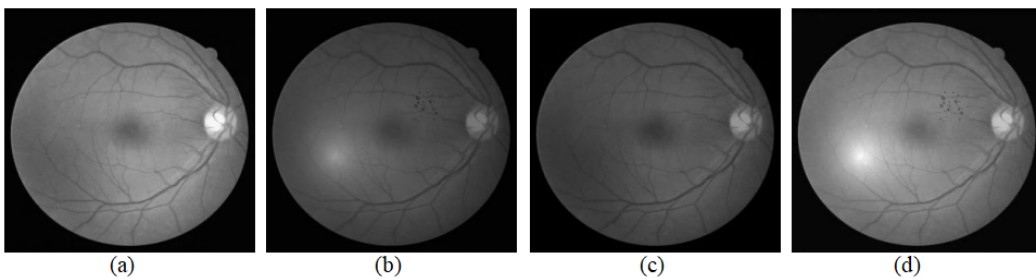

|   (a)   |   (b)   |   (c)   |   (d)   |

**Figure 1.** Intensity normalization of the image pair. The original image taken as the reference image is from DRIVE. A small noise patch is attached on the reference image and the intensity is adjusted which is taken as the current image. (**a**,**b**) are the grayscale images of the reference image and current image, and (**c**,**d**) are results of intensity normalization for (**a**,**b**).

In fact, after intensity correction and normalization, the comparison of $\tilde{I}_1$ and $\tilde{I}_2$ is converted into the comparison of two image pairs at two different intensity levels: the image pair $\hat{I}_1$, $\hat{I}_{21}$ and the image pair $\hat{I}_2$, $\hat{I}_{12}$. In order to filter the distraction of illumination variations and keep the background more robust, we take the linear interpolation to obtain more comparisons of the image pair on different intensity levels.

*2.2. Linear Interpolation*

For statistical background modeling, to obtain a stable background model, sufficient sampling frames are required [19]. If the sequence itself contains only a few frames of images, it is usually difficult to generate a stable background model, as the intensity abruptness in the sequence will significantly distract the model. In order to obtain more background frames to produce a stable background model, linear interpolation is used in two different intensity levels of the image pair to expand the short sequence into a long sequence with slowly changing intensity [11].

For two images, $I_1$ and $I_2$, supposing the reference image $I_1$ is taken in the early stage, which can be regarded as the background image, then the current image $I_2$ is used to compare with the background to discriminate whether there was a change. Since $I_1$ is turned into $\tilde{I}_1$ and $\hat{I}_{21}$ after taking the illumination correction and normalization [10], the low-rank background model is based on $I_1$, so linear interpolation is performed between $\tilde{I}_1$ and $\hat{I}_{12}$. The formula is presented in (3):

$$\bar{I}_{1k} = \frac{k}{p_1}\hat{I}_{12} + (1 - \frac{k}{p_1})\tilde{I}_1, \quad k = 1, 2, \cdots, p_1, \tag{3}$$

where $\bar{I}_{1k}$ is the $k$-th interpolated frame $k = 1, 2, \cdots, p_1$, and $p_1$ is the number of interpolated frames. Generally, the larger the $p_1$, the smaller the intensity abruptness between the two successive frames is, and the more stable the background model is.

Apart from the linear interpolation between the reference frames, it is also required for the current frame. For the current frame $I_2$, the linear interpolation between $\tilde{I}_2$ and $\hat{I}_{21}$ is shown in the formula (4):

$$\bar{I}_{2k} = \frac{k}{p_2}\hat{I}_{21} + (1 - \frac{k}{p_2})\tilde{I}_2, \quad k = 1, 2, \cdots, p_2 \tag{4}$$

where $p_2$ is the number of interpolated frames of $I_2$.

After the linear interpolation of the reference image and the current image at their different brightness levels, the influence of illumination variations on the change regions is further reduced. Then, it can be easily filtered out as a low-frequency component by the detection algorithm [10]. Linear interpolation makes the comparison of the two image pairs with different intensity levels extend to the comparison of multiple different intensity levels, which enabled the abruptness between different intensity levels to decompose to the small global illumination variations of two successive frames and be filtered as the small disturbed errors of the background model by low-rank decomposition.

## 3. Methodology

Anatomic structures captured from the same eye generally evolve slowly as time elapses, which can be regarded as the principal low-rank background component of the longitudinal image serial [10]. Lesions often occupy only a few pixels in the image that change with time, which corresponds to a temporal highpass filter. Therefore, longitudinal fundus images are modeled as the low-rank sequence and change detection is obtained by the low-rank decomposition. The background modeling approach takes the whole image as the background and considers the more spatial neighborhood of anatomic structures than the pixel-by-pixel method; hence, better detection results and clear change regions can be obtained.

### 3.1. Low-Rank Decomposition Modeling

Suppose the longitudinal fundus images have $N$ frames, $I_1, I_2, \cdots, I_N$, and each frame has the size $m \times n$, $N \geq 2$. Vectorizing each frame as a vector, these longitudinal fundus images can be regarded as an image matrix with the size $M \times N$ denoted by $D$ where $M = m \times n$ and $D = [I_1, I_2, \cdots, I_N]$. Each column of $D$ indicates one frame of the sequence, and hence $D$ consists of the following two components in (5):

$$D = A + E. \tag{5}$$

where $A$ is the low-rank background component of $D$, $E$ is the sparse change component of $D$. $A_i$ and $E_i$ correspond to the background image and the change region image of $I_i$.

The change detection problem can be converted into how to obtain the sparse component of $D$, which can be solved by low-rank decomposition as the following equation shows:

$$\arg \min_{A,E} rank(A) + \lambda \|E\|_0 \quad s.t. \quad D = A + E \tag{6}$$

where $rank(A)$ indicates the rank of matrix $A$, $\|\cdot\|_0$ means the $l_0$ norm which shows the sparsity of the component, and $\lambda$ is a hyperparameter and the equilibrium factor to trade off the rank of the image matrix and sparsity of the foreground.

As solving the $l_0$ norm in (6) is a non-deterministic polynomial (NP) problem [26], according to restricted isotropic property (RIP) [27], the $l_0$ norm can be estimated by the $l_1$ norm. The rank of the matrix is measured by the kernel norm of matrix, and (6) can be reformulated to the form of (7), shown as follows [28]:

$$\arg \min_{A,E} \|A\|_* + \lambda \|E\|_1 \quad s.t. \quad D = A + E, \tag{7}$$

where $\|\cdot\|_*$ is the kernel norm, and $\|\cdot\|_1$ is $l_1$ norm.

Many studies [29,30] have given the solution of (7). Since there are some noises involved in the image for the general cases, (7) can be relaxed from the equality constraint to the inequality constraint to allow some errors, which is shown in (8):

$$\arg \min_{A,E} \|A\|_* + \lambda \|E\|_1 \quad subj \quad \|D - A - E\|_F^2 < \varepsilon \tag{8}$$

The augmented Lagrange multiplier (ALM) algorithm [31] gives a fast solution method shown in (9):

$$\arg \min_{A,E} \|A\|_* + \lambda \|E\|_1 + \frac{\mu}{2} \|D - A - E\|_F^2 \tag{9}$$

where $\|\cdot\|_F$ is the Frobenius norm of the matrix, and $\mu$ is a parameter used to balance the reconstruction accuracy as well as the low-rank and sparsity of the image matrix.

### 3.2. Patch Low-Rank Decomposition Modeling

In (6), the whole frame is taken as one column of the image matrix, and the great similarity of the columns makes the image sequence become a one-rank matrix. For the P-RPCA method, the parameter $t$ is set as the length of each square window which means that the window has a size of $t \times t$. For the image of size $m \times n$, it is divided into many patches by the window of size $t \times t$, and the number of patches in each frame is given by (10).

$$T = [\frac{m}{t}] \times [\frac{n}{t}] \tag{10}$$

where $T$ is the number of the patches and $[\cdot]$ indicates a rounded-up operation.

If the size of the image cannot be exactly divided, we should pad the boundary with a gray value of 0 so that the size can be exactly divided. The size of the image can also be adjusted by scaling the image. The segmented image patch sequences are concatenated together successively in the time dimension. Figure 2 illustrates the division

and concatenation. Hence, the original sequence with $N$ frames is turned into an image sequence with $T \times N$ frames.

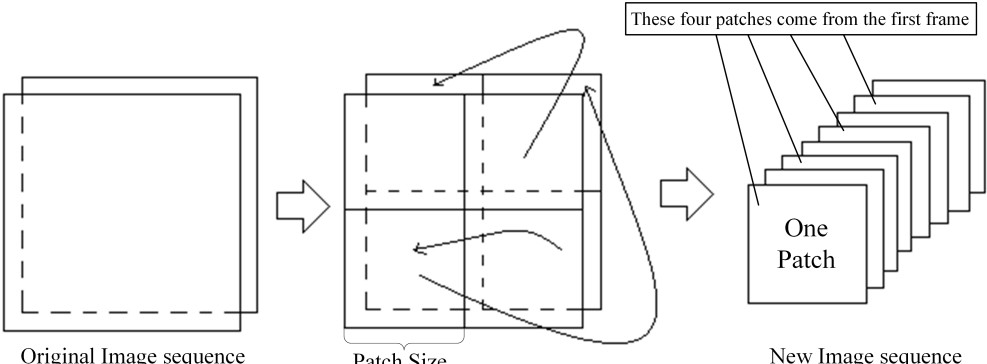

Original Image sequence      Patch Size      New Image sequence

**Figure 2.** Procedure of obtaining the image patches sequence. The image here is divided into four patches with $N$ frames. These images are concatenated to obtain a new image sequence with $T \times N$ frames.

Similarly, each frame of the new image sequence is vectorized as a column, and then the sequence is converted into an image matrix denoted by $D'$ with a size of $t^2 \times TN$. Then, the low-rank decomposition can be used to obtain the sparse change regions of the sequence. The augmented Lagrange multiplier (ALM) algorithm [31] can also give the solution method of the optimization problem in (11):

$$\arg \min_{A',E'} \|A'\|_* + \lambda \|E'\|_1 + \frac{\mu}{2} \|D' - A' - E'\|_F^2 \tag{11}$$

Once the low-rank component $A'$ and the sparse component $E'$ of $D'$ are obtained, one can recover the background of the original image and sparse change regions according to original stacking order. On the one hand, P-RPCA turns the local illumination variations inside the image into that between the successive frames which can be filtered by the low-rank decomposition into the reconstruction error. Hence, compared with RPCA, P-RPCA is better at dealing with local illumination variations [10].

On the other hand, the rank of the image patch matrix becomes higher than the original image matrix since the patches in the same image are strongly dissimilar, which makes increases rank after division and concatenation. As the number of the consistent background becomes at least the number of patches, the rank of the image matrix increased to $T$ from rank one. For the optimization of (11), the kernel norm, the first term on the right side of the equation, becomes significantly bigger than before, which forces the second term of sparsity and the third term of the reconstruction error to be smaller. Hence, P-RPCA can reconstruct a more accurate background and clear change regions at the same time than RPCA.

*3.3. Algorithm and Flow Chart*

This section illustrates the steps by Figure 3 and presents the basic algorithm procedure of change detection proposed in this paper, as shown in Algorithm 1.

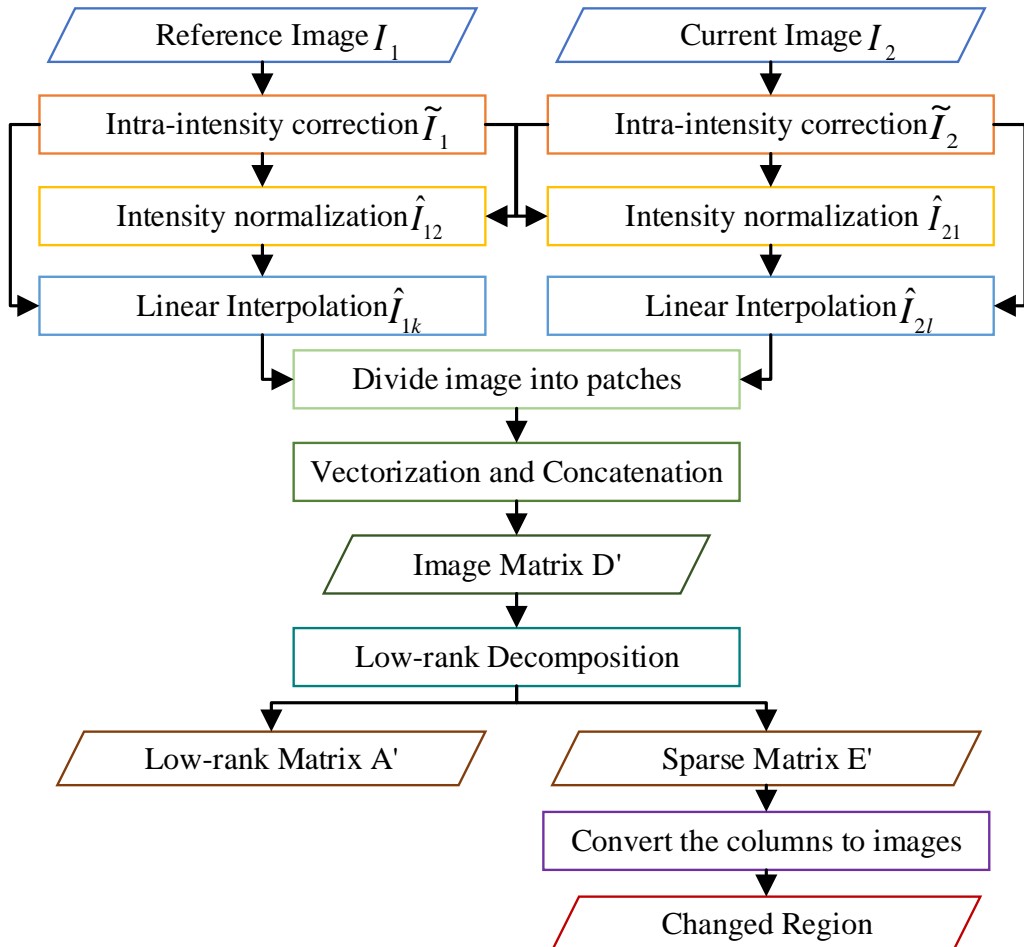

**Figure 3.** Flow chart of P-RPCA.

---

**Algorithm 1** Algorithm procedure of P-RPCA

---

**Input:** A registered pair of fundus image $I_1$ and $I_2$;

**Output:** Change detection of $I_2$ according to $I_1$

1: Correct the intra-image intensity of $I_1$ and $I_2$ to get $\tilde{I}_1$ and $\tilde{I}_2$;

2: Normalize the intensity of $\tilde{I}_1$ and $\tilde{I}_2$ according to (1) and (2) to obtain $\hat{I}_{12}$ and $\hat{I}_{21}$;

3: Interpolate $p_1$ frame between $\tilde{I}_1$ and $\hat{I}_{12}$ according (3), interpolate $p_2$ frame between $\tilde{I}_2$ and $\hat{I}_{21}$ according (4);

4: Vectorize $\hat{I}_{1k}$ and $\hat{I}_{2l}$, $k = 1, 2, \cdots, p_1$, $l = 1, 2, \cdots, p_2$, divide the image sequence into patch image sequences and concatenate them into $D'$;

5: Do low-rank decomposition on $D'$ to get low-rank background matrix $A'$ and sparse foreground matrix $E'$;

6: Convert the columns of $E'$ back to images, and they are recovered to the size of the original image according to the original stacking order, the final image sequence can be represented as $I_E$, $I_E = \{I_{E1}, I_{E2}, \cdots, I_{EN}\}$, and take the frames from $I_E$ as the change component.

---

## 4. Experiments and Discussion

In this experiment, $\lambda$ is the reciprocal of the size of a single image block after dividing, $\mu$ is set to 1.25 times the reciprocal of the maximum singular value of $D$. The registration in this paper is based on the local partial intensity invariant feature descriptor (PIIFD) [32]. In the modeling of a low-rank image sequence, there are 20 interpolated background images and 3 foreground images.

### 4.1. Data

The experiments validated the effectiveness of the proposed method from two pairs of fundus images with different lesion sizes. Figure 4 shows these two pairs of fundus images.

Figure 4a,b are the fundus image pair from the public dataset DRIVE [25]. Figure 4a is the original image taken as the reference image of the image pair, and Figure 4b is one attached with the red noise patch regarded as small lesions on Figure 4a, which is seen as the current image. Figure 4c,d are the images corrected by IRHSF and illumination models are presented in Figure 4e,f separately.

Figure 4g,h are the fundus image pair from Xinhua Hospital in Shanghai. The original image pair is of high resolution and is under-sampled to $403 \times 384$. Figure 4g is the reference image with a slight random spot on the low-right corner. Figure 4h is the current image involved with a big bright lesion. The image pair is also enhanced and corrected by IRHSF. Their corrected images are presented in Figure 4i,j separately, and their illumination models are shown in Figure 4k,l, respectively.

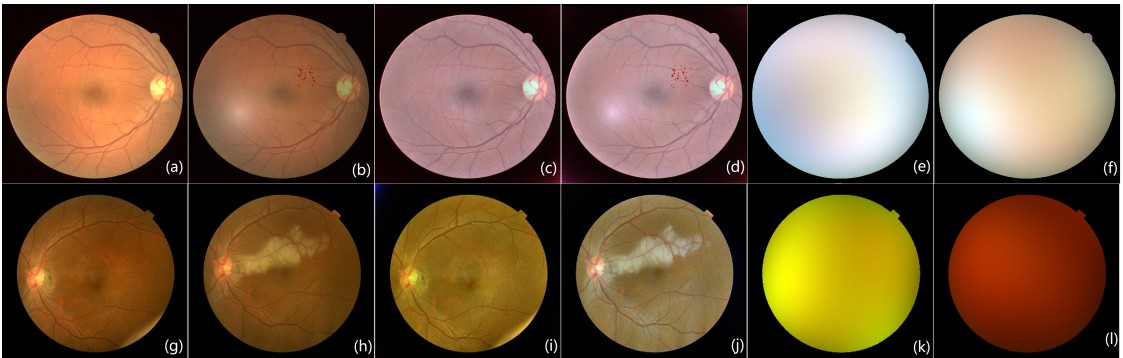

**Figure 4.** Fundus image pairs with a different size of lesions. (**a**) Reference image from DRIVE; (**b**) current image attached with the small noise patch; (**c**) corrected images of (**a**) by IRHSF; (**d**) corrected images of (**b**) by IRHSF; (**e**) illumination model of (**a**); (**f**) illumination model of (**b**). (**g**,**h**) are from Xinhua Hospital in Shanghai. (**g**) Reference image, (**h**) current image, (**i**) corrected images of (**g**) by IRHSF, (**j**) corrected images of (**h**) by IRHSF, (**k**) illumination model of (**g**), (**l**) illumination model of (**h**).

### 4.2. Validation Measurement

The proposed method was validated by ROC curve and PR curve. The AUC designed to evaluate the comprehensive performance of the classifier is calculated through the ROC curve where TPR and FPR denote the vertical and horizontal coordinates separately. MAP is used to calculate the average accuracy value through the PR curve where precision and recall denote the vertical and horizontal coordinates, respectively. The four indexes are given by the following formulas:

$$TPR = \frac{TP}{TP + FN} \tag{12}$$

$$FPR = \frac{FP}{TN + FP} \tag{13}$$

$$Precision = \frac{TP}{TP + FP} \tag{14}$$

$$Recall = \frac{TP}{TP + FN} \tag{15}$$

where $TP$, $FP$, $TN$, and $FN$ indicate true positive, false positive, true negative, and false negative, respectively.

### 4.3. Results with P-RPCA Method

For P-RPCA, strictly dividing the image may not be applicable for change regions of different sizes. Hence, a sliding window with overlap is a more flexible technique than the strict division for the change detection. The P-RPCA with a sliding window is taken to improve the performance of the proposed method. After the size and stride of the sliding window are set, the slide window sweeps across the first frame of the sequence and produces many overlap subsequences with the same size. By concatenating all the subsequences together and vectorizing them, an patch image matrix is obtained. By the sliding window, the content of many patches may be redundant, which is useful to produce a consistent low-rank background and filter the local light spots well [18].

The algorithm is practiced on two groups of data shown in Figure 4. The image pair with small lesions has a resolution of $604 \times 584$ as shown in Figure 4a,b. Figure 1 presents the registered and normalized image pairs. Figure 5 gives the detection result of the low-rank decomposition by RPCA and P-RPCA after normalization and linear interpolation. Figure 5c,d are the reconstructed low-rank background by RPCA and P-RPCA, respectively, and Figure 5e,f are their sparse change regions given by RPCA and P-RPCA, respectively. The red circle in Figure 5e marks the local flare distraction.

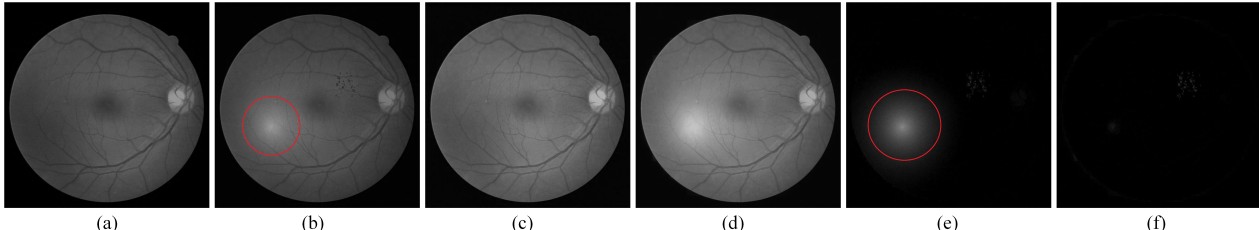

(a)    (b)    (c)    (d)    (e)    (f)

**Figure 5.** The result of the image with small lesions by RPCA method. (**a**) Normalized reference image; (**b**) normalized current image; (**c**) reconstructed low-rank background by RPCA; (**d**) reconstructed low-rank background by P-RPCA; (**e**) sparse change regions by RPCA; and (**f**) sparse change regions by P-RPCA. The red circle marks the distraction of flare in the current image for the RPCA method.

For the image pair with small lesions, as shown in Figure 5, P-RPCA with a sliding window is used to detect the change regions. Through several attempts to choose different sizes of equal intervals for the experiments, the appropriate size of the sliding window was set to be 200 and the stride was set to be 50. It can be seen that compared with the detected change features in Figure 5e,f, these are less distracted by the local light spot and the detected change features are clearer. The red mark in Figure 5e is caused by the local flare. Furthermore, in Figure 5f, this distraction is greatly reduced. The change regions obtained by P-RPCA method obtain less distractions by the local flare and are clearer and cleaner than those by RPCA.

The image pair with a big lesion has a resolution of $403 \times 384$. In Figure 6, (a) and (b) are the normalized reference image and the current image, respectively, after registration, and (c) and (d) are the reconstructed background image by RPCA and P-RPCA, respectively. Figure 6e is the result obtained by the image matrix and low-rank decomposition after linear interpolation. The image pair from Figure 4g,h is corrected by IRHSF and normalized to each other, and then it is expanded to a sequence. For the reference image, the number of interpolated frames is 20 in order to obtain a stable background, and for the current image, the number of interpolated frames is 3. The image sequence after interpolation includes 23 frames and each image frame has a size of $403 \times 384$. The concatenated image matrix $D$ has a size of $154752 \times 23$. It can be seen that the features of the isolated large lesions are

obvious, but the light spot in the reference image significantly distracts the change region as the red circle marks.

For the image pair with large lesions, padding the boundary of the image to let the size be $600 \times 400$ and linearly interpolating the image pair to be of 23 frames. Then, the size of patch is set to be 200 and divide the image into six patches. The image sequence is turned into 138 frames, and the concatenated image matrix $D'$ is of size $40{,}000 \times 138$. Figure 6 presents the results of a low-rank decomposition on $D'$. Compared with the detected change features in Figure 6, the result in Figure 6 obtained by P-RPCA is clearer and cleaner, which reduces the distraction of illumination change and camera position movement.

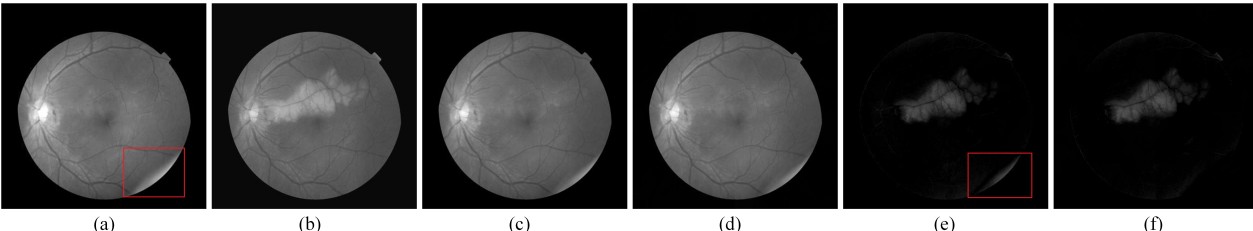

(a)      (b)      (c)      (d)      (e)      (f)

**Figure 6.** The result of the image with large lesions by RPCA method. (**a**) Normalized reference image; (**b**) normalized current image; (**c**) reconstructed low-rank background by RPCA; (**d**) reconstructed low-rank background by P-RPCA; (**e**) sparse change region by RPCA; and (**f**) sparse change region by P-RPCA. The red circle marks the distraction of the light spot in the reference image for the RPCA method.

### 4.4. Discussion

The proposed method is validated from the quantitative evaluation on the ROC curve and PR curve in this section. The ground truth of the change detection is given in Figure 7. The ground truth is artificially marked. The result of the P-RPCA method in the following curves is obtained with the appropriate size by experiments.

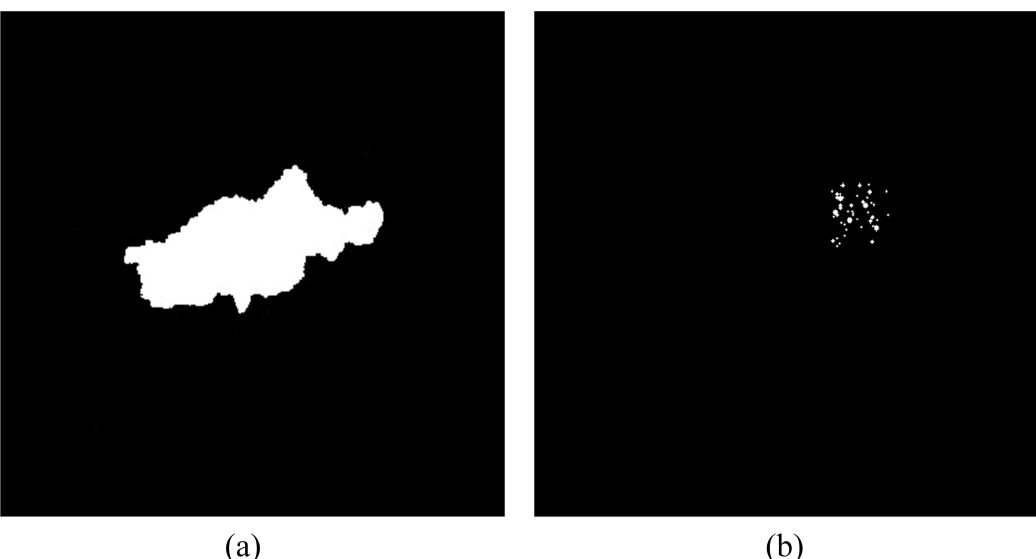

(a)                  (b)

**Figure 7.** Ground truth of the image pair with small lesions and big lesions. (**a**) Big lesions; and (**b**) small lesions.

The curves from the image pair with small lesions and a big lesion are given in Figures 8 and 9 separately. The four indexes are calculated through statistical analysis of each pixel in the image. Among them, true positive means detecting the correct positive sample, and false positive means the false detection is a true negative sample, and so on.

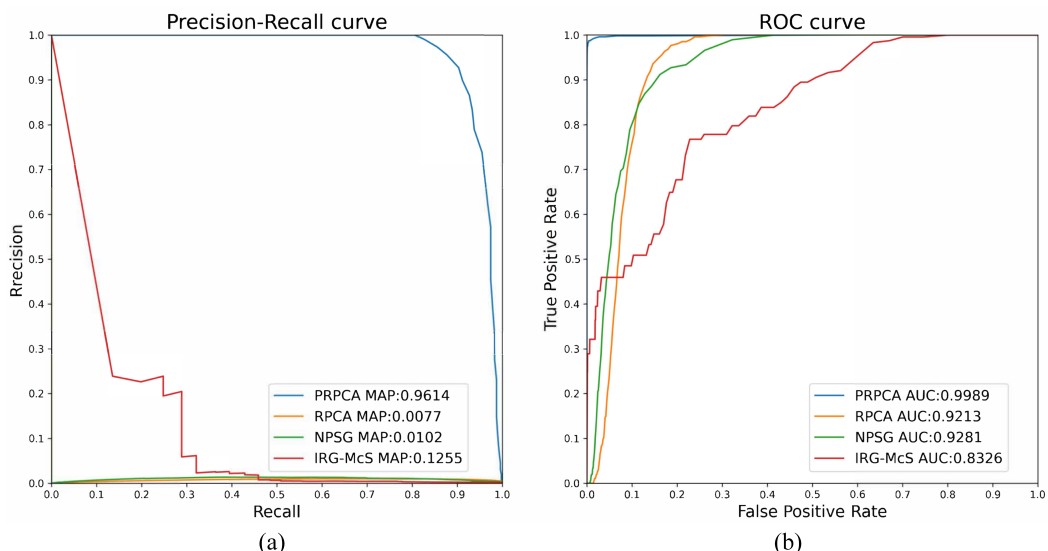

(a)

(b)

**Figure 8.** ROC and PR curve results of the image pair with small lesions. (**a**) PR curve; and (**b**) ROC curve.

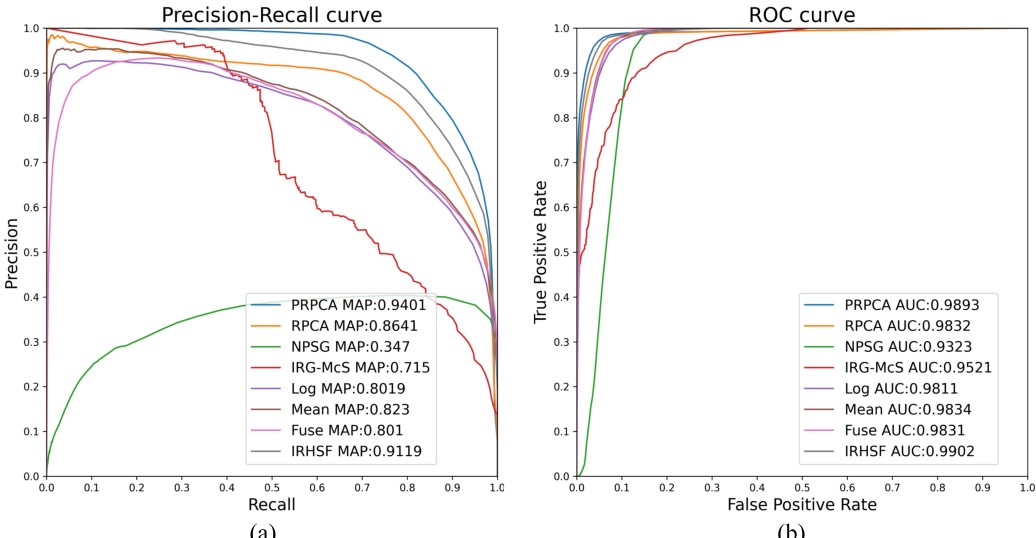

(a)

(b)

**Figure 9.** ROC and PR curve results of the image pair with a large lesion. (**a**) PR curve; and (**b**) ROC curve.

For the image pair with small lesions, four change detection techniques are used to prove the performance of the proposed methods, as illustrated in Figure 8: RPCA, IRG-McS [33], NPSG [34], and P-RPCA. The RPCA method performs poorly in the face of local flare and the result is significantly distracted, whilst the AUC and mAP are 0.9213 and 0.0077, respectively, which is far lower than 0.9989 and 0.9614 for P-RPCA. In Figure 8, both the ROC and PR of P-RPCA are over the curves of the other three methods, which means that P-RPCA is significantly better than the other methods.

For the image pair with the large lesion, five change detection techniques from the remote sensing are introduced here: IRHSF difference [11]; mean difference [16]; logarithmic quotient [16]; wavelet fusion [16]; IRG-McS [33]; and NPSG [34]. As shown in Figure 9, the AUC and mAP are 0.9832 and 0.8641, respectively, for RPCA, while 0.9893 and 0.9401 for P-RPCA. As can be seen from the curve in Figure 9, the performance of the P-RPCA method exceeds RPCA method on almost all thresholds, and far exceeds other algorithms.

Figure 10 presents the detection results of the different method under the image pair with small lesions shown in Figure 4. For RPCA, the local flare significantly distracts the

change regions and so does IRG-McS as shown in Figure 10b,d. IRG-McS presents the obvious change regions as the red rectangle marks but is very sensitive to the other facts such as the intensity of the optic disc. P-RPCA is obviously better than the other three methods as Figure 10 illustrates.

Figure 11 presents the detection results of a different method for the image pair with the large lesion shown in Figure 4. Compared with Figure 11e–g, Figure 11a,b are more robust to the local illumination variations and less affected by the vascular tissue. In Figure 11d, the result is evidently significantly distracted by the boundary . The result in Figure 11a is clearer and cleaner than in Figure 11b and less distracted by the light spot in the reference image.

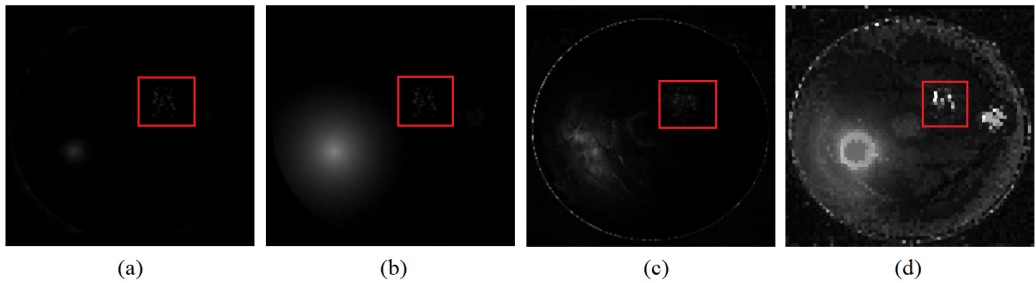

(a)      (b)      (c)      (d)

**Figure 10.** Detected change region for the image pair with small lesions in Figure 4. (**a**) P-RPCA; (**b**) RPCA; (**c**) NPSG; and (**d**) IRG-McS. The red rectangle marks the small change regions.

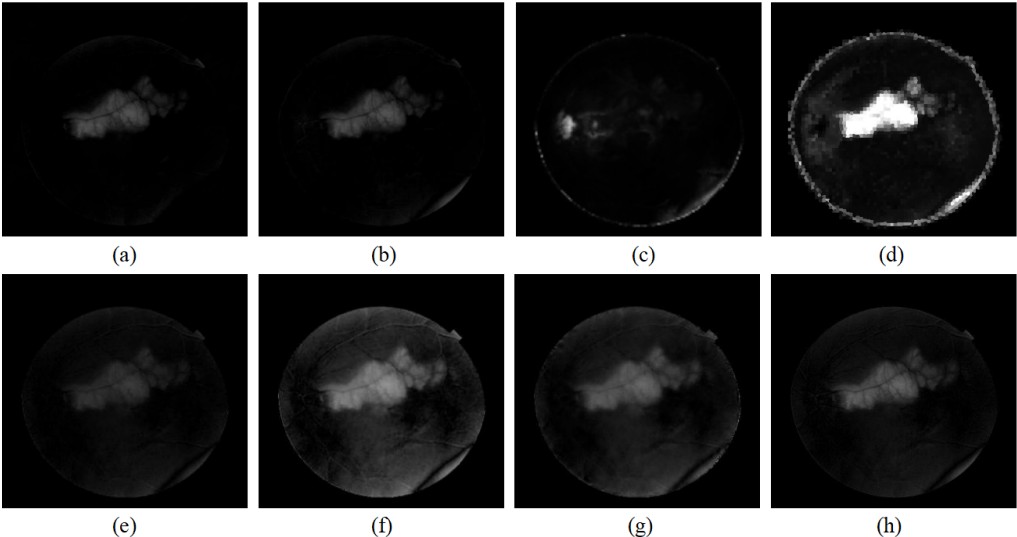

(a)      (b)      (c)      (d)

(e)      (f)      (g)      (h)

**Figure 11.** Detected change region for the image pair with the big lesion in Figure 4. (**a**) P-RPCA; (**b**) RPCA; (**c**) NPSG; (**d**) IRG-McS; (**e**) logarithmic quotient; (**f**) mean difference; (**g**) wavelet fusion; and (**h**) IRHSF difference.

Figure 12 shows the influence of IRHSF on AUC and mAP values. The dataset used in the experiment is the image pair with small lesions. When intensity correction methods are not used, the AUC and mAP are 0.9981 and 0.957, respectively. After using intensity correction methods, such as IRHSF, the AUC and mAP are 0.9989 and 0.9614, respectively. This shows that after using the intensity correction method to correct global illumination variations for the image pair, change regions detected by P-RPCA method are more accurate.

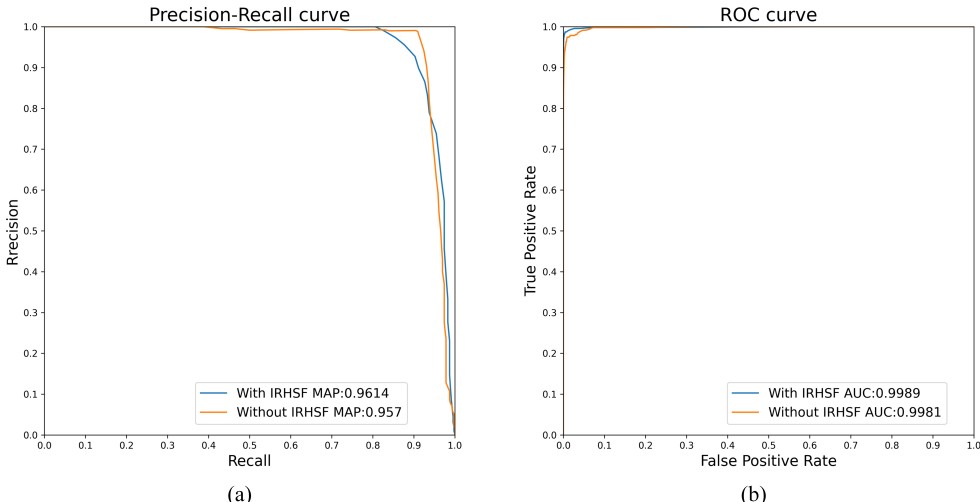

(a)　　　　　　　　　　　　　　　　(b)

**Figure 12.** Influence of IRHSF on AUC and mAP values. (**a**) PR curve, (**b**) ROC curve.

Figure 13 shows the influence of patch size on AUC and mAP values in the P-RPCA method. The dataset used in the experiment is the image pair with small lesions. The blue line, the orange line, the green line, and the red line indicate that the size of the patch is taken as 50, 100, 200, and 300, respectively. When the size of the patch is taken as 200, the AUC and mAP obtain the maximum value, which is 0.9989 and 0.9614, respectively. Figure 14a–d show the change detection results of the image pair when the patch size in the P-RPCA method is 50, 100, 200, and 300, respectively, and the detection results are shown in red circles. Hence, for the image pair with small lesions, the size of the patch selected by the P-RPCA method is 200.

As the size of the sliding window will affect the result of change detection, we repeat the experiment by changing the size to obtain the best value for different image pairs. If the sliding window is too small, it is difficult for the method to detect the change regions. If the sliding window is too big, the method will only weakly reduce the distraction due to local illumination variations.

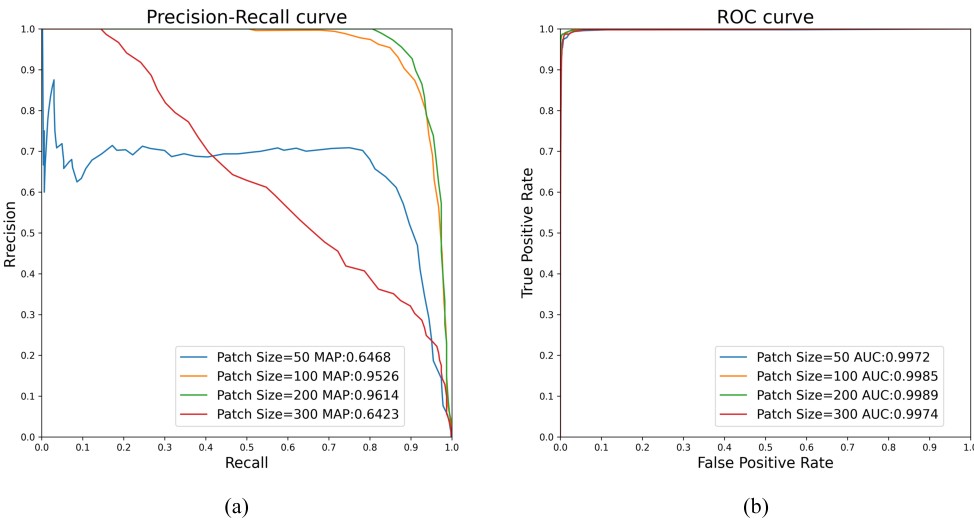

(a)　　　　　　　　　　　　　　　　(b)

**Figure 13.** Influence of patch size on AUC and mAP values in the P-RPCA method. (**a**) PR curve, (**b**) ROC curve.

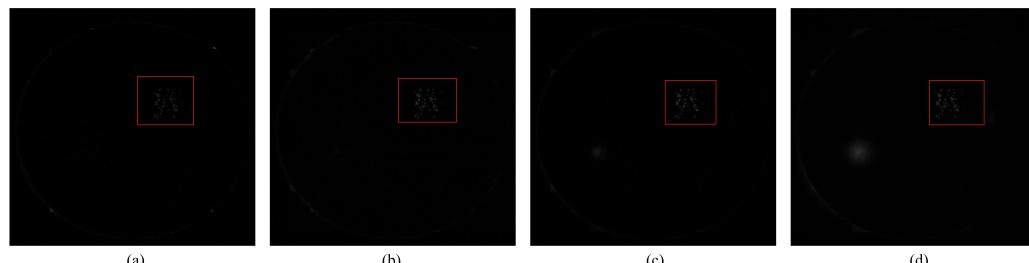

| (a) | (b) | (c) | (d) |

**Figure 14.** Influence of patch size on detection results in the P-RPCA method for the image pair with small lesions. (**a**) Patch size = 50; (**b**) patch size = 100; (**c**) patch size = 200; and (**d**) patch size = 300.

Generally, when there are fewer frames in the sequence, the background of the sequence is more susceptible to illumination variations since the single background frame is complicated to correct to the variable illumination variations of the current frames. Linear interpolation reduces the global illumination variation between the successive frames and makes the intensity slowly change in the sequence. The global illumination variation becomes smooth so that it can be filtered as the reconstruction error when the sequence is decomposed into a low-rank component and the sparse component, which is the advantage of RPCA. P-RPCA further reduces the distraction of illumination variation by dividing the sequence into several subsequences and decomposes the concatenated subsequences which is more robust than RPCA.

## 5. Conclusions

For the change detection of the retinal fundus image pair, the method based on P-RPCA is proposed in this paper which is the improved version to matrix RPCA. Decomposing the image matrix by patches can deal with both the global and local illumination and is more robust for the distraction of the illumination variations than RPCA and obtains clearer change regions. The main advantage of P-RPCA lies in that P-RPCA can reduce the distraction of the local flare or random light spot to a certain extent further. Choosing the appropriate size of patches to divide the sequence is an important factor to determine the performance of change detection. How to choose the patch size and stride is the next research work in the future.

**Author Contributions:** Conceptualization, Y.F.; Data curation, X.Z.; Formal analysis, Z.Z.; Funding acquisition, Y.F.; Methodology, Y.F.; Project administration, Y.F.; Software, W.Z.; Validation, W.Z. and X.Z.; Visualization, Z.Z.; Writing—original draft, W.Z.; Writing—review & editing, Z.Z., X.Z. and Y.F. All authors have read and agreed to the published version of the manuscript.

**Funding:** This research was funded by Open Project Foundation of Intelligent Information Processing Key Laboratory of Shanxi Province OF funder grant number CICIP2021003.

**Institutional Review Board Statement:** Not applicable.

**Informed Consent Statement:** Not applicable.

**Data Availability Statement:** Not applicable.

**Conflicts of Interest:** The authors declare no conflict of interest.

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
