# Peer review of "Changed Detection Based on Patch Robust Principal Component Analysis"

_applsci, doi:10.3390/app12157713_

Round 1

Reviewer 1 Report

General comment:

This work deals with the use of patch-based RPCA for analyzing the variations in fundus image pairs, reducing the influence of local illumination.  

Specific comments throughout the paper:

Abstract. 

In the abstract no references should be used. Please revise the abstract according to the authors' guidelines: 

https://www.mdpi.com/journal/applsci/instructions

Line 12: The use of non defined acronyms (AUC, mAP) does not allow to get a clear and immediate comprehension. Please revise.

1. Introduction

Line 21: plese revise the reference numbering after removing the [1] from the abstract. 

Lines 29-30: Missing references. I suggest to read 10.1109/RBME.2010.2084567 and 10.1016/j.preteyeres.2005.07.001 to expand and support this part. 

Lines 62-71: The literature gap and the novelty of the proposed approach must be better specified and discussed.

"Figure 1. Intensity normalization of the image pair. The original image taken as the reference image

is from DRIVE." - Please report the information about DRIVE. 

For Eq. (1) and (2) there is no comparison with other methodology, thus a quantitative demonstration of the value and validity of the adjustmenet has not been provided. 

2. Intensity Correction and Interpolation

Line 130: Missing refrence for the LF filter. 

3. Change Detection Model

Lines 137-139: Missing reference(s).

In Eq. (6) the term labda is not defined. Please define all symbols. 

Line 156: Define the acronym for the sake of clarity. 

Fig. 2 is not so clear to me. Please revise it to get a better figure. 

The formulation of the problem appear to be robust. 

Fig. 3: There is no trace in the manuscipt of spare matrix S'. Probably this is an error and E should be reported in this figure. Please check. 

4. Experiments and Discussion

Line 208: Please provide the reference for the dataset. This is not fair to the readership. 

Line 213: Ethical statements? Additional details are needed. 

A suitable, robust and thorough validation was not performed. A deep and coherent discussion about this point must be provided. 

Eqs. (12)-(15): There is mixture of methods and results. Please re-consider the paper organization to get a clear and understandable manuscript. 

5. Conclusion 

The conclusion section can be enhanced. 

------------------------------------------------------

Minor edits:

Line 27: "As the change is of diversity" - Please check and control the english language. It is not clear. 

Line 37: "many researchers take the ..." - not appropriate language. 

Line 124: repeated p1. Please fix 

Line 166: do not intend after equation. 

Reviewer 2 Report

Change detection on retinal fundus image pairs is mainly to compare the important differences between a pair of images obtained at two different time points such as anatomical structures or lesions. Illumination variation usually challenges the change detection methods in many cases. Robust principal component analysis (RPCA) takes intensity normalization and linear interpolation to reduce greatly the illumination variation between the continuous frames and then decomposes the image matrix to obtain the robust background model [1]. Matrix-RPCA can obtain clear change regions but still is distracted by local bright spots and the background model is still vulnerable for the illumination.

In this paper, patch-based RPCA (P-RPCA) is proposed by the authors  to detect the change of fundus image pairs, where a pair of fundus images is normalized and interpolated linearly to expand a low-rank image sequence, then images are divided into many patches to obtain an image-patch matrix, finally the change regions are obtained by the low-rank decomposition.

The authors  validated their methodology on a set of large lesion image pairs in clinical data.

The authors concludes that their results show that RPCA method is more robust to local illumination changes than RPCA method, and has stronger performance in change detection than RPCA method.

The manuscript is interesting and well written.

Some minor suggestions for the authors:

1.      Correct the abstract. For example the reference citation  [1] must be deleted

2.      Rearrange the ms according to the standards: insert the methods and a clear purpose

3.      Insert the limitations in the discussion

Reviewer 3 Report

In this work, authors have presented patch based RCA approach to detect changes between a pair of fundus images. The work is interesting, however, there are a few points that need to be addressed.

1. Authors need to proof read the manuscript carefully especially on sentence formation and run-on sentences.

2. Authors have not presented the results of the intensity correction in a quantitative manner. Please provide a plot of the intensity matrix to evaluate the comparison in a quantitative manner.

3.It is not clear how the optimal patch size was determined?

4. Authors have not presented ground truth details.

5. How does the patch size affect AUC ?

6.What was the split of training and test data?

7. What is the source of the images? and were the images randomly selected for testing?

8. How does this model compare with previous models for change detection?

Round 2

Reviewer 1 Report

The authors improved the referencing. 

Thanks for Lines 73-80. 

The validation, the parameter selection and the results presentation are now more clear and their quality significantly improved.

The fact that some information are provided in the discussion is acceptable.  

Reviewer 3 Report

Authors have appropriately addressed most of the comments.